# ARCFIRE: Experimentation with the Recursive InterNetwork Architecture

**Sander Vrijders [1], Dimitri Staessens [1], Didier Colle [1], Eduard Grasa [2,*], Miquel Tarzan [2], Sven van der Meer [3], Marco Capitani [4], Vincenzo Maffione [4], Diego Lopez [5], Lou Chitkushev [6] and John Day [6]**

1. imec IDLAB, Ghent University, B-9000 Ghent, Belgium; sander.vrijders@ugent.be (S.V.); dimitri.staessens@gmail.com (D.S.); didier.colle@ugent.be (D.C.)
2. Fundació i2CAT, Software Networks Research Area, 08034 Barcelona, Spain; miquel.tarzan@i2cat.net
3. Ericsson Network Management Lab, N37 Athlone, Ireland; sven.van.der.meer@ericsson.com
4. Nextworks s.r.l, Knowledge Services, 56122 Pisa, Italy; capitani.mrc@gmail.com (M.C.); v.maffione@gmail.com (V.M.)
5. Telefónica Investigación y Desarrollo S.A, Technology Exploration, 28050 Madrid, Spain; diego.r.lopez@telefonica.com
6. Metropolitan College, Computer Science, Boston University, Boston, MA 02215, USA; ltc@bu.edu (L.C.); day@bu.edu (J.D.)
* Correspondence: eduard.grasa@i2cat.net

**Abstract:** European funded research into the Recursive Inter-Network Architecture (RINA) started with IRATI, which developed an initial prototype implementation for OS/Linux. IRATI was quickly succeeded by the PRISTINE project, which developed different policies, each tailored to specific use cases. Both projects were development-driven, where most experimentation was limited to unit testing and smaller scale integration testing. In order to assess the viability of RINA as an alternative to current network technologies, larger scale experimental deployments are needed. The opportunity arose for a project that shifted focus from development towards experimentation, leveraging Europe's investment in Future Internet Research and Experimentation (FIRE+) infrastructures. The ARCFIRE project took this next step, developing a user-friendly framework for automating RINA experiments. This paper reports and discusses the implications of the experimental results achieved by the ARCFIRE project, using open source RINA implementations deployed on FIRE+ Testbeds. Experiments analyze the properties of RINA relevant to fast network recovery, network renumbering, Quality of Service, distributed mobility management, and network management. Results highlight RINA properties that can greatly simplify the deployment and management of real-world networks; hence, the next steps should be focused on addressing very specific use cases with complete network RINA-based networking solutions that can be transferred to the market.

**Keywords:** RINA; experimentation; Quality of Service; resiliency; renumbering; mobility; network management

---

## 1. Introduction

RINA—the Recursive InterNetwork Architecture—starts from the premise that networking is "Inter-Process Communication (IPC) and IPC only" [1]. Networks provide the means by which applications on separate systems communicate, generalizing the model of local IPC. In contrast to the fixed, five-layer model of the internet, where each layer provides a different function, RINA proposes a single configurable type of layer—called a Distributed IPC Facility (DIF)—that provides an IPC service to the layer(s) above. The IPC service is defined by the IPC Abstract Service Definition, which defines

the operations that the DIF provides to (i) allocate resources between applications, called "flows"; (ii) read and write data from these flows; (iii) de-allocate the flows and free the resources associated to them. In RINA, invariant parts (mechanisms) and variant parts (policies) are separated, making it possible to customize the behavior of a DIF with sets of policies tailored to a particular environment. RINA recognizes that no single set of policies can be effective over the entire range of requirements a network may encounter. DIFs can be stacked as many times as necessary and each one can be configured to support a given range of bandwidth and Quality of Service (QoS) requirements.

Three research initiatives have been previously funded by the European Commission into researching RINA: FP7 IRATI (Investigating RINA as an Alternative to TCP/IP) [2] developed the first RINA implementation over ethernet for the Linux Operating System. FP7 PRISTINE (Programmability in RINA for European supremacy on virtualized networks) [3] built on IRATI's efforts to further improve the prototype and start applying RINA to specific areas such as congestion control, resource allocation, routing, security and network management. The GN3+ open call winner IRINA (Investigating RINA as the next generation GEANT and NREN architecture) [4] researched the potential for RINA in the framework of GEANT and National Research and Education Networks (NRENs).

Following these three projects, there was a need for a more experiment-driven project that leverages the prototypes and the European investment in Future Internet research infrastructures. The main objectives of the H2020 ARCFIRE (Large-scale RINA benchmark on FIRE) project were to:

(1) Compare the design of converged operator networks using RINA to state-of-the-art operator network designs.
(2) Produce a robust RINA software suite; mature enough for large-scale lab experiments.
(3) Provide relevant experimental evidence of the RINA benefits for network operators, application developers, and end-users.
(4) Raise the number of organizations involved in RINA research, development, and innovation activities.
(5) Enhance FIRE+ as a platform for large-scale experimentation with RINA. FIRE+ (Future Internet Research and Experimentation) is a European experimental infrastructure for Future Internet research.

We show how a converged operator network can be modelled using RINA in Section 2; how ARCFIRE increased the scale of testbed experimentation with the RINA prototypes with some experimental results in Section 3; and how RINA impacts network management in Section 4. Section 5 provides a discussion of the implications of the experimental results presented in this paper. The article is concluded in Section 6.

## 2. Modeling a Converged Network Operator

ARCFIRE has analyzed how a converged operator network (CON) with an access-metro-core architecture can be modelled with RINA [5], considering the following steps:

- Create the overall layered structure. The network is partitioned into a set of inter-operable layers (DIFs), each with a certain scope. The lower layers focus on resource allocation and QoS of more deterministic traffic, while the higher layers focus on grooming traffic to effectively use the lower layers. A trade-off has to be taken into account to decide the optimal number (typically no more than 5), since having more layers will have the effect of bounding the router table size [6], yet it also means an increase in delay and more packet headers.
- Characterize the QoS requirements for each layer. Each network is designed to serve a set of applications with specific traffic characteristics over a set of physical media. The way in which the network is broken up into layers as well as the role of each layer will define a set of QoS characteristics that need to be supported by each layer. Recognizing that these systems are not sufficiently sensitive to be configured to a precise point in "QoSspace", i.e., precise set of values

for the QoS parameters, it is defined in terms of one or more "QoS-cubes". Network designers can map regions in this performance space to the QoS-cubes to be supported by each layer.

- Characterize the security requirements of each layer, in terms of authentication, access control, confidentiality or integrity protection.
- Specify the exact packet header syntax in the layer, based on the expected number of concurrent flows, the number of addressable IPC Processes, the maximum size of payload (larger packet sizes towards the backbone), the number of QoS cubes, etc.
- Decide per-layer policies, which allow the layer to operate effectively and meet the quality requirements (expressed via the QoS cubes).
- Identify operating ranges for the network and define monitors/triggers to sense them and choose/design the most appropriate strategies to automatically transition in the Network Management System (NMS) between different policy sets for each layer associated to the identified network operating ranges.

Figure 1 illustrates the Converged Operator Network use case considered by ARCFIRE. It considers different types of access networks—xDSL (Digital Subscriber Loop), FTTH (Fibre to the Home), WiFi and cellular—that allow the provider to reach its customers (residential, business, mobile) via wired and wireless media. These technologies are chosen due to their high representability, but other access networks could be considered (e.g., Hybrid Fiber/Coax networks) without modifying the design. The traffic from these access networks is aggregated by metropolitan area networks (which may cover a neighborhood, a city or a semi-urban/rural area depending on the density of customers), which aggregate the traffic towards the core. The traffic is then forwarded towards another core Point of Presence (PoP)—if it is addressed to another customer of the service provider or to a service hosted in the service provider's DataCenters (DCs)—or to an Interconnect edge router (e.g., internet edge) otherwise. The service provider may have datacenters attached to different parts of the network: MicroDCs attached to access networks in support of Mobile Edge Computing; Metro DCs attached to the metropolitan networks to support Content Delivery and cloud computing services; and large regional DCs attached to the core networks.

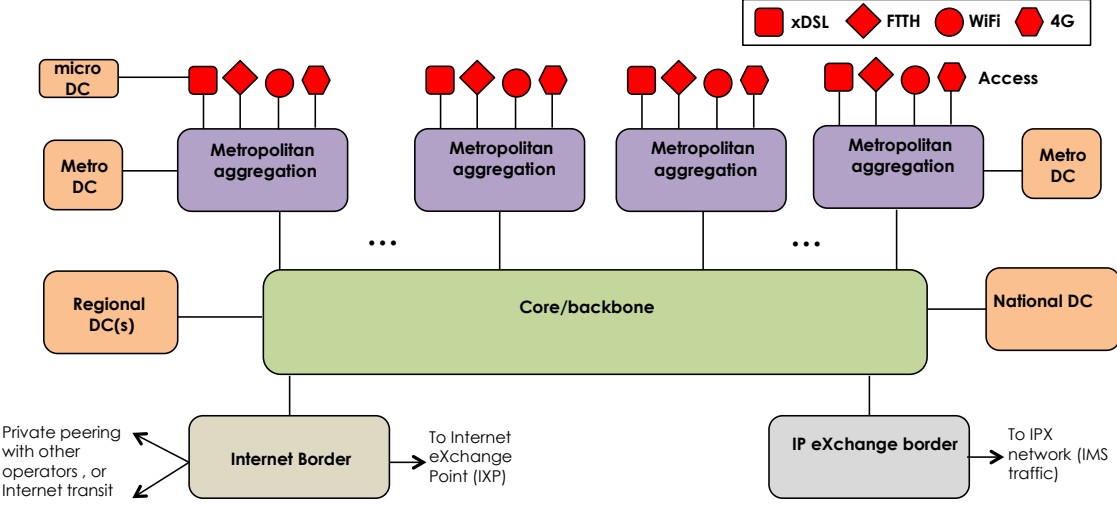

**Figure 1.** High-level view of a Converged Operator Network (CON) as considered by ARCFIRE.

Figure 2 shows a potential DIF structure for the CON in the case of a residential customer accessing remote applications over a top-level DIF (public internet DIF, Virtual Private Network or VPN, application-specific, network slice, etc.). The upper part of the figure shows the connectivity graph of the network boxes, while the bottom part illustrates the different DIFs going through such boxes. The metropolitan segment—consisting in Metropolitan Area Networks (MANs)—aggregates

the traffic of different service DIFs coming from access routers from multiple types of access networks. This segment can be modelled with one or two DIFs that multiplex traffic and deliver it to service routers in core Points of Presence. The residential customer DIF can be thought of as an internal network slice that manages the resources that the operator has allocated to residential customers through the different segments of the network (access, metro, core). Other types of customers (e.g., businesses, different verticals) may have their own separate service DIFs. The backbone (BB) segment connects together service routers in different metropolitan regions, as well as provider border routers that interconnect with other providers directly or via internet eXchange Points.

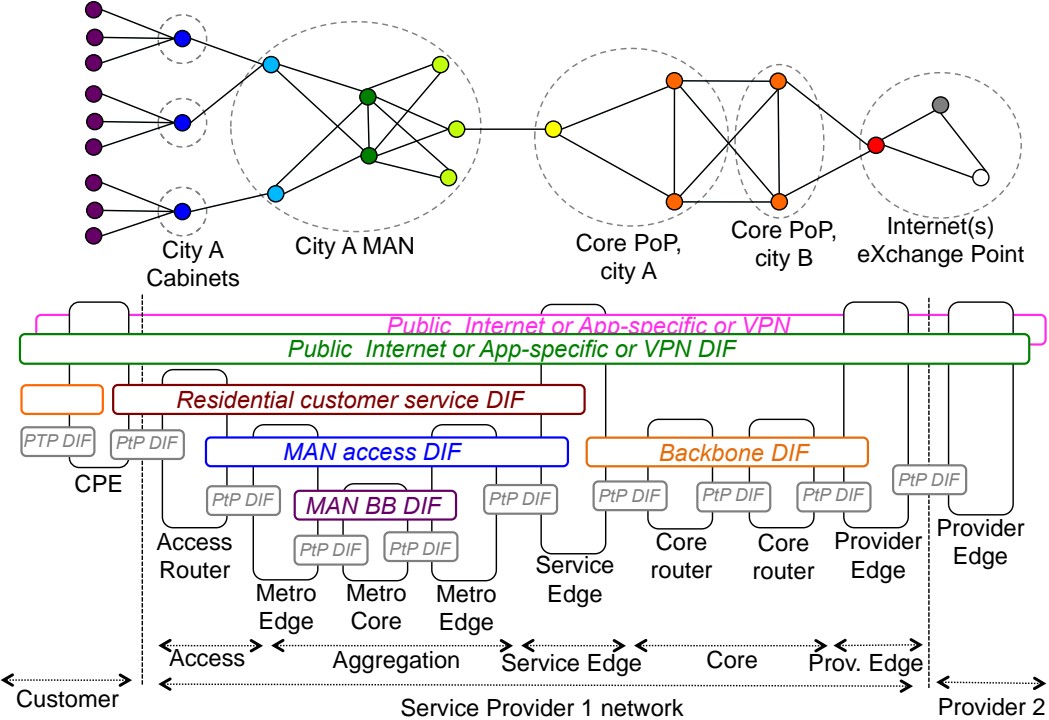

**Figure 2.** Example design for a CON: fixed residential access.

The backbone can consist of one or more DIFs, depending on the scale of the CON core network. Access routers connect to the Customer Premises Equipment (CPEs) directly or via an access network segment, depending on the access technology. Figure 2 depicts a customer connected via a direct fiber from the access router (modelled as a point-to-point DIF). Figure 3 illustrates the cellular access scenario. The major changes with respect to Figure 2 are the introduction of Cell DIFs between Mobile Hosts (MHs) and Base Stations and the presence of the Serving Area DIF to manage mobility within a set of neighboring cells. More levels of DIFs can be employed to accommodate the load, scale, and rate of change of the mobile devices to be supported [7]. A full description of the RINA design for a converged operator network—including DIF structures and policy proposals for each segment of the network—is provided in Section 3 of ARCFIRE's Deliverable D2.2 report [5].

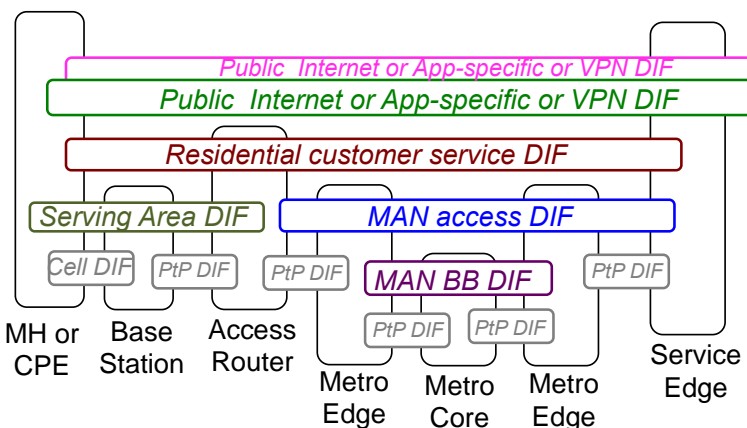

**Figure 3.** Example DIF design for a Converged Operator Network: cellular access.

Table 1 shows a summary of architectural properties exhibited by a converged operator network designed using RINA compared to the ones shown by a CON designed with current state of the art networking technologies as fully described in ARFIRE's Deliverable D2.1 report [8].

**Table 1.** Comparison of architectural properties exhibited by RINA converged operator networks and CONs built using the current network protocol architecture.

| CON Design | with Current Network Architectures | with RINA |
|---|---|---|
| *Structure* | Different layers, different functions | Different layers, same functions with different policies |
| | Functions in different layers are not always independent | Strict layering: layers are black boxes, only the service interface is visible |
| | Fixed and static number of layers, require additional constructs such as tunnelling, virtualization, etc. | Variable number of layers, decided at network design or operation time |
| | Incomplete and/or missing layer service definitions | Consistent and common service definition across layers |
| | Unclear division of functions at each layer, "control" and "data" planes | Precise and consistent structure of the functions within each layer |
| *Protocol Design* | Multiple protocols per layer, protocol proliferation | Only two protocol frameworks per layer, configurable via policies |
| | Protocols independently designed from each other: high variability | Common functions in all layers, only policies can change: simplicity |
| *Naming, addressing, and routing* | Lack of application names: the network uses IP addresses and port numbers to identify applications | Complete naming and addressing architecture, simplifies mobility, and multi-homing of applications |
| | Naming the interface instead of the node, making transparent multi-homing and mobility hard to achieve | Addresses assigned to nodes, simplifies multi-homing, and mobility of network connected devices |
| | Network renumbering causes downtime | Transparent network renumbering |
| | No names for layers, no layer directory | Naming DIFs enables application discovery across layers |
| *Mobility and multi-homing* | Mobility and multi-homing limited and requires specialized protocols | Mobility and multi-homing achieved via constructs in the architecture |

**Table 1.** *Cont.*

| CON Design | with Current Network Architectures | with RINA |
|---|---|---|
| *Resource allocation and Quality of Service (QoS)* | Lack of consistent QoS model, each protocol family has its own | Consistent QoS model across layers, from application to physical layers |
| | Only end-to-end congestion control loops—reaction time is maximized | Congestion control loops shorter, closer where congestion occurs |
| | Implicit congestion detection in many cases—false positives and overreaction | Explicit Congestion Detection always, for a precise response |
| | Homogeneous congestion control policies for heterogeneous networks | Per-layer congestion control policies: optimal reaction to congestion in different parts of the network |
| *Security* | Most protocol enforce their own security model, introducing complexity | Secure layers instead of individual protocols: consistent security model |
| | Use of well-known ports facilitates transport layer attacks | Use of application names, dynamic transport port allocation |
| | Network addresses exposed to applications, facilitates network scanning | Addresses are internal to a layer and never exposed outside of it |
| *Network Management* | Too little commonality in network protocols, making management of CONs increasingly complex | Two immutable protocol frameworks and a well-defined set of policies simplify the network structure |
| | No well-defined model for the interaction between layers | Consistent service definition and function structuring across layers |
| | Different protocols and object modelling frameworks for network management | Single network management protocol and common layer object model |

The usage of the same building block (the DIF) configured with different policies simplifies network design, and makes it more flexible: more layers can be introduced dynamically by the network operator to help the network scale, optimize the use of resources and properly isolate different scopes (such as different customers or network segments). Small networks can start with a small number of DIFs, in order to minimize the protocol and management overhead, and scale by introducing new ones in different parts of the network when needed. Applications are not restricted to use a specific top-level DIF, they can use whatever DIF provides the scope and quality that fulfils the application requirements. The network operator can exploit this ability to provide optimized DIFs that let customers communicate at lower layers, with the benefit of increased performance. For example, customers that are physically close to each other could communicate directly via a DIF that directly sits on top of the access network layers. Or the operator can leverage this ability to provide Mobile Edge Computing (MEC) services via DIFs that give access to latency-critical applications hosted in micro-DCs at the edge of the network.

A RINA-based CON uses a single management protocol, and all DIFs have the same base object model that captures that DIF's common behavior. Individual policies provide information about their state and configuration by extending that base model. RINA networks feature a converged model and northbound management interface for any network segment (access, aggregation, core, interconnect); thus, facilitating the re-use of operations support systems (OSS) applications for fault, configuration, security, accounting and performance management. In comparison, current OSS systems are usually tied to a specific vendor, equipment model or software version. A full discussion of these architectural properties is provided in Section 3.8 of ARCFIRE's Deliverable D2.2 [5].

### 3. Scaling up RINA Networks

Before the start of ARCFIRE, the IRATI prototype supported short-lived tests with minor traffic variations on a relatively small number of systems. Two tools were available to validate the implementation: the configurator and the demonstrator [9]. The demonstrator allows a user to define and run a RINA network on local QEMU Virtual Machines (VMs) using configuration files. IRATI had to be installed before running the demonstrator. The configurator allowed defining RINA networks through Extensible Markup Language (XML) files, and would run this on physical machines in an EMULAB testbed. The configurator starts the basic daemons of IRATI on the machines but more advanced configuration was not supported.

These tools showed it was necessary to automate the execution of large experiments, since doing this manually is error-prone, hard to reproduce, and very time consuming. To fill this gap, ARCFIRE developed Rumba [10]. Rumba is an experimentation framework that supports existing testbeds provided by the American GENI (Global Environment for Network Innovation) and the European FIRE initiatives via the jFed and Emulab experiment management software. Rumba provides a simple yet powerful API for experimenters to configure all the aspects of a RINA network, deploy it, run it on testbeds, and gather the results. It is implemented as a Python library that allows the user to programmatically define (i) the physical connectivity graph of the network; (ii) how RINA layers are laid out on the nodes, without any restriction on layer membership and stacking geometry; (iii) the policies to be used by each layer; (iv) where and when distributed applications should run. Rumba already supports several testbeds and prototypes and is easily extendable. Rumba has been used by all experiments carried out within ARCFIRE, and helped further test scalability of the supported prototypes. At the end of the projects all supported prototypes can scale up to hundreds of nodes, thousands of IPCPs (IPC Processes, the individual instances of DIFs in a system) and ten-thousand flows.

#### 3.1. Resiliency within a Single DIF

If a failure occurs in the current internet, there is no unified mechanism to signal a failure between layers, which hampers fast recovery. RINA has the concept of a flow, which can be marked as down upon failure. We implemented a Loop-Free Alternates (LFA) [11] policy for RINA, which has the goal to reduce failure reaction time to tens of milliseconds by using a pre-computed alternate next-hop that can be used as soon as a failure of the primary next-hop is detected. We deployed a network that uses the physical connectivity graph of the GÉANT network on FIRE using Rumba. As depicted in Figure 4, the minimum degree of the graph is 2, the maximum 5 and the average degree is 2.93. Each link is implemented on the testbed as an ethernet DIF and a normal DIF is run on top of them. In total, the experiment has 30 nodes and 44 links, so there are 30 normal IPCPs, and 88 ethernet IPCPs.

An application sending data at a constant bit rate has been deployed on the "exterior" nodes of the network, i.e., those geographically far from the core of the continent. The nodes selected as "end-nodes" are Ankara, Dublin, Lisbon, Ljubljana, Moscow, Nicosia, Oslo, Riga, Stockholm, and Valleta. Several links have been selected to represent faulty links: every 30 s one of them would go down for 15 s and come back up. Several 50 Mbps connections are created, emulating a realistic load on the net. Figure 5 shows the running average over 1 s of the RTT (Round Trip Time) of several of the paths starting from Ljubljana for a total duration of 270 s. The link down and link up events are marked with vertical lines. We can see that around those times, some of the paths suddenly increase or decrease their RTT, as some flows change their states to down, forcing the normal IPCP to reroute packets via an alternate path. The recovery is almost instantaneous, it takes around 15 ms to recover, well below the carrier grade recovery requirement of 50 ms.

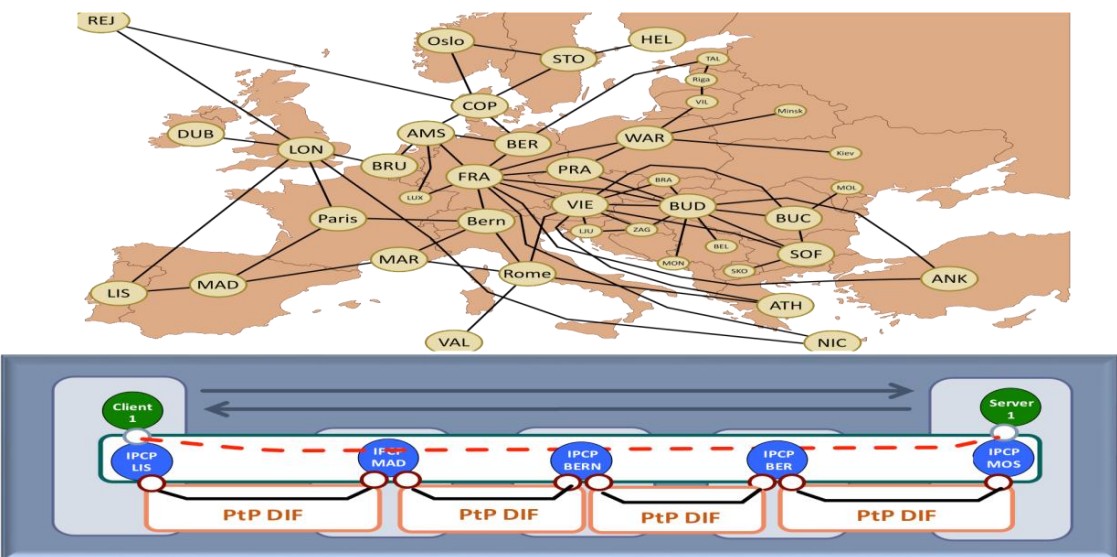

**Figure 4.** GEANT Network physical connectivity graph and DIF structure used in the experiment.

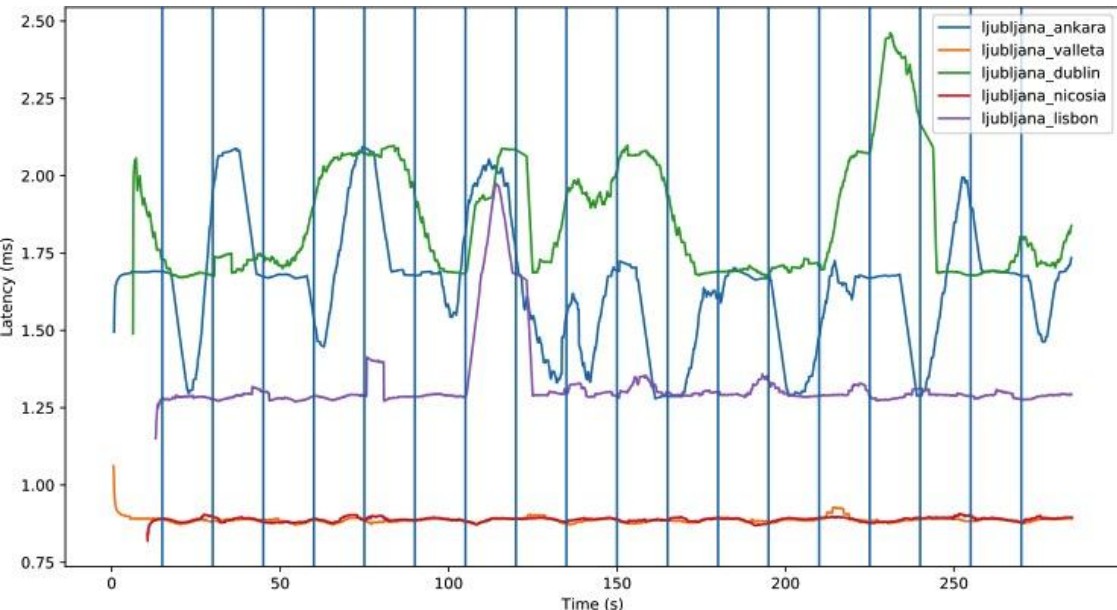

**Figure 5.** RTT (Round Trip Time) for some of the paths involving Ljubljana.

### 3.2. Renumbering RINA Networks

Renumbering IP networks is an expensive procedure that has to be carefully planned and executed to avoid routing, firewall, and connection integrity problems [12]. TCP connections are tightly bound to a pair of IP addresses, and will be disrupted with changing IP addresses. Furthermore, the renumbering process usually leads to stale DNS (Domain Naming System) entries pointing to deprecated addresses [13].

The use of application names and directories per DIF makes renumbering a RINA network easier. Applications request flows to target application names, which are resolved using the DIF directory to the address of an IPC Process (IPCP). Renumbering assigns a new address (synonym) to the IPCP, giving it 2 addresses. The IPCP sends a message for all its flows that originate at this IPCP to its peer. This notifies the other end of the flow so that it can use the new address. A routing update in the IPCP starts advertising the new address instead of the old one. The DIF's directory is updated so that

applications are now mapped to the new address. Over time, all peers will use the new address and the old address disappears.

ARCFIRE renumbering experiments showed that it is possible to dynamically renumber RINA networks (even multiple DIFs simultaneously), without destroying any flows and with a minimal impact on the QoS experienced by each flow [14]. Initial experimental evidence of this property was already described in [12], but with very limited experimental setup consisting in few flows and systems running in Virtual Machines. ARCFIRE performed a multi-layer renumbering experiment on the imec iLab.t Virtual Wall testbed with 41 physical machines and 6 DIFs, as shown in Figure 6. Each IPCP in each DIF changed its address once every 45 s, while 180 flows between CPE routers were exchanging traffic using a simple ping application. We performed an experiment with and without network renumbering. There was no packet loss or observable change in RTT, as shown in Figure 7.

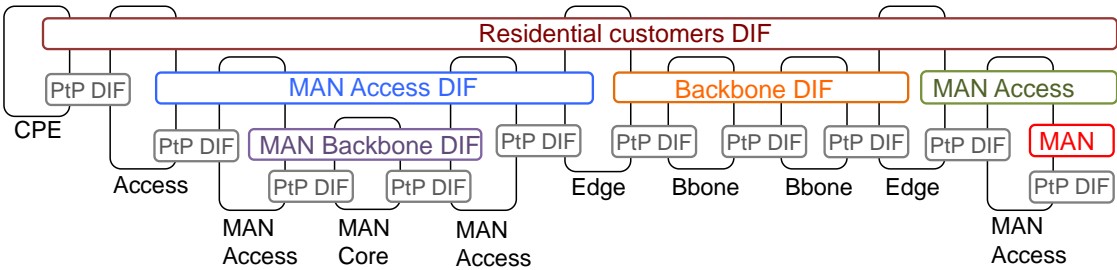

**Figure 6.** DIF configurations for the renumbering experiments.

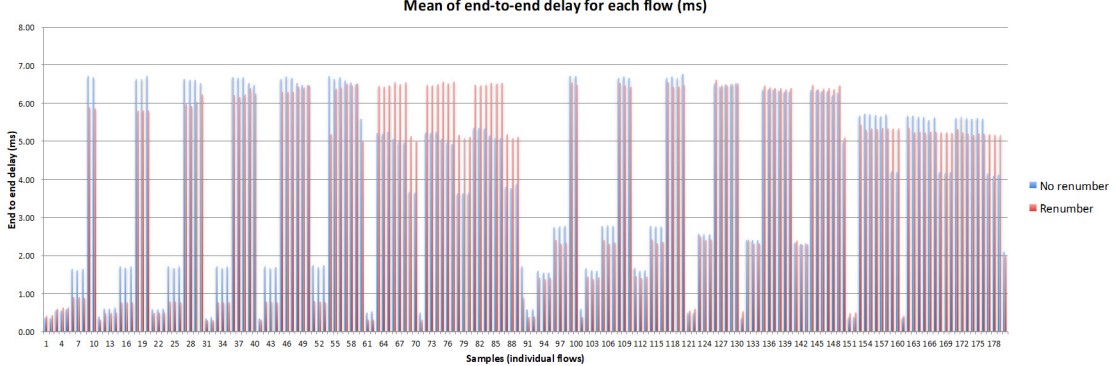

**Figure 7.** Application RTT, measured with and without renumbering, for each flow.

In this experiment we analyzed how the complete naming and addressing architecture embodied by RINA allows RINA networks to be renumbered live, without significantly impacting the performance perceived by existing flows or impairing the ability to crate new ones. Renumbering a network with multiple layers today is a maintenance event: it has to be carefully planned, requires humans in the loop, and it takes one or more days to complete. With RINA renumbering can be done life, dynamically, in a fully automated way, and in a time period between tens or hundreds of milliseconds to a few seconds (depending on the network size).

### 3.3. Providing QoS Guarantees

The RINA flow allocation interface allows applications (including IPCPs) to specify the quality requirements on a flow in a technology-agnostic way. When an application requests a flow from a layer, the requested quality for a flow is first mapped to a defined QoS cube (internal to the layer), and then the most appropriate transport and resource allocation policies are selected to meet the application requirements. Each packet sent by an IPCP carries a QoS cube identifier that clearly identifies packets belonging to each QoS cube, and enables resource allocation policies (routing, scheduling, congestion control) to act consistently across a DIF.

Technologies such as MPLS or Segment Routing [15] provide strong guarantees on the QoS demands but rely on dedicated paths (tunnels), source routing, or incur in considerable protocol header overhead. RINA can provide a more distributed approach to QoS in order to address use cases such as provider-based IP/ethernet VPNs, data-center fabrics or networks slicing. A DIF cannot only use virtual circuits and static allocation to support QoS, but any combination of routing, scheduling, and congestion control policies that enforce the service level defined by each QoS cube.

ARCFIRE carried out experiments emulating a RINA-based IP VPN provider that offers IP services on top of a RINA network, illustrated by Figure 8. The symmetric scenario consists of 4 core nodes in a ring, each connected to 3 provider edge (PE) routers (12 in total). Each PE router provides IP connectivity to 2 customer edge (CE) routers. The experiment established 6 IP VPNs, with 4 CE routers at each IP VPN. The core DIF uses Quantitative Transport Agreement (QTA) resource allocation policies [16] to support four classes of service (QoS cubes) with differential loss and delay: urgent, low loss and urgent, high loss and non-urgent, low loss and best effort. Figure 9 shows the Cumulative Distribution Functions (CDF) of the end-to-end delay experienced by flows between PE routers, for medium- and high-load scenarios in two configurations: one in which the DIF uses FIFO (First In First Out) schedulers, and the other QTA schedulers. During medium load the offered traffic is about 50% of the DIF's capacity, and during high load 80%. An application sending data at a constant bit rate was used. The QTA resource allocation policies enable the core DIF to guarantee an upper bound in the end-to-end delay for the most urgent QoS cubes, at the cost of degrading the delay of the less urgent ones.

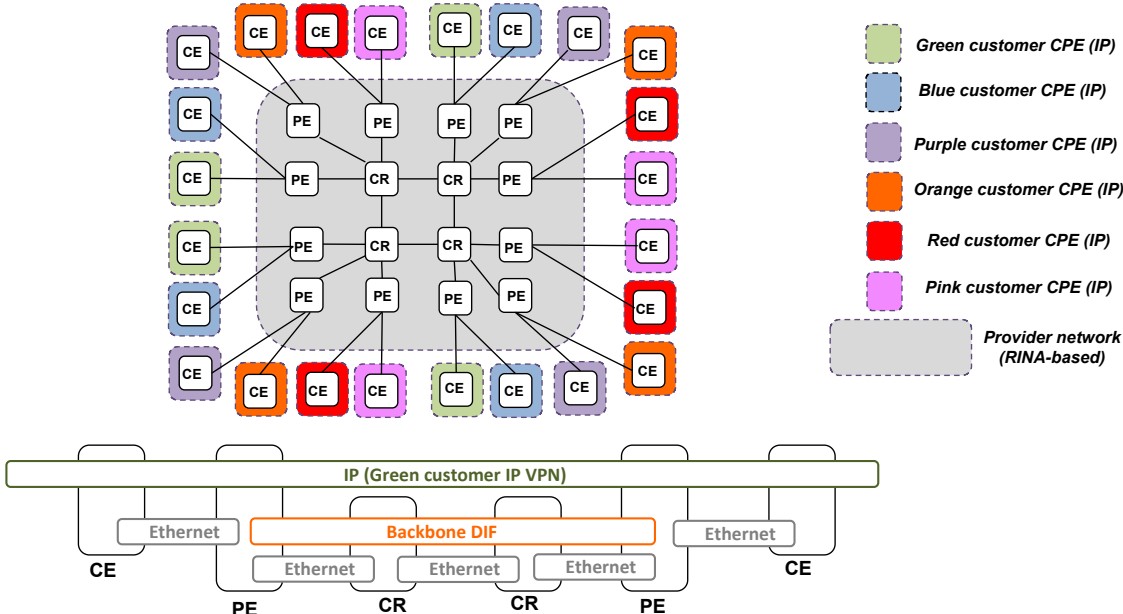

**Figure 8.** Systems and DIFs in the RINA-based QoS-enabled Virtual Private Network experiment.

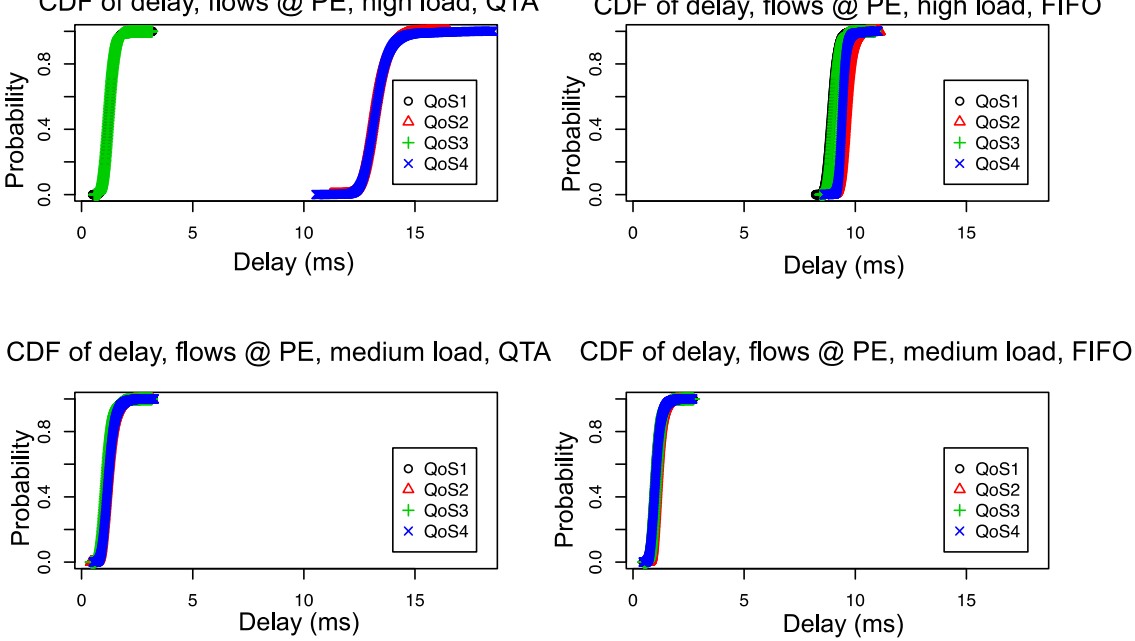

**Figure 9.** End-to-end delay Cumulative Distribution Function (CDF) for flows between PEs, using QTAMux (**left**) and FIFO (**right**) schedulers in the DIF. QoS1 and QoS3 are the urgent classes.

For low or medium loads (lower part of Figure 9), the behavior of both policies is almost equivalent: flows belonging to different QoS cube experience a similar end-to-end delay and there is no packet loss. The differences in the behavior of both policies become very clear at high loads (higher part of Figure 9). In the QTA policy case, flows belonging to urgent QoS cubes experience a similar end-to-end delay than in the low load scenario (with an average value of 1.25 ms), while flows belonging to non-urgent classes experience an end-to-end delay that is 10 times higher (with an average value of 13.25 ms). There is also some packet loss experience by the non-urgent flows belonging to the less-cherished QoS class (QoS cube 4). In the FIFO policy case, all QoS classes experience an end-to-end delay much higher than the urgent classes of the QTA policy (9.85 ms), but significantly lower than the non-urgent classes. The results reflect the nature of the trade-off: the QTA policy assigns more scheduling capacity to urgent classes at the cost of degrading the quality experienced by non-urgent classes, increasing its delay and packet loss compared to the non-differential treatment case (FIFO policy). The scheduling capacity of the IPCP is constant, the QTA policy allows network designers and administrators to decide how this scheduling capacity can be allocated through different QoS classes within a RINA DIF.

We performed a second experiment in a multi-layer scenario, to validate RINA's consistent QoS model across layers. The physical systems in the experiment setup are shown in Figure 10, with 41 nodes featuring a provider network that connects together 18 CPE routers belonging to two different metropolitan regions, via a core network. Each metropolitan region features a Metropolitan Area Network (MAN) that aggregates the traffic of all the CPEs in the region towards the core network. The DIF structure is the same as displayed by Figure 6. Only the residential customer DIF is equipped with QoS differentiation capabilities and the QTA scheduling policies, all the other DIFs just feature simple FIFO scheduling policies. The goal of this scenario was to increase the scale of the former QoS scenario experiments, in terms of the number of RINA-enabled nodes (from 16 to 41), DIFs (from 1 to 6) and flows (from 128 to 1296), as well as to validate the behavior in a multi-layer environment.

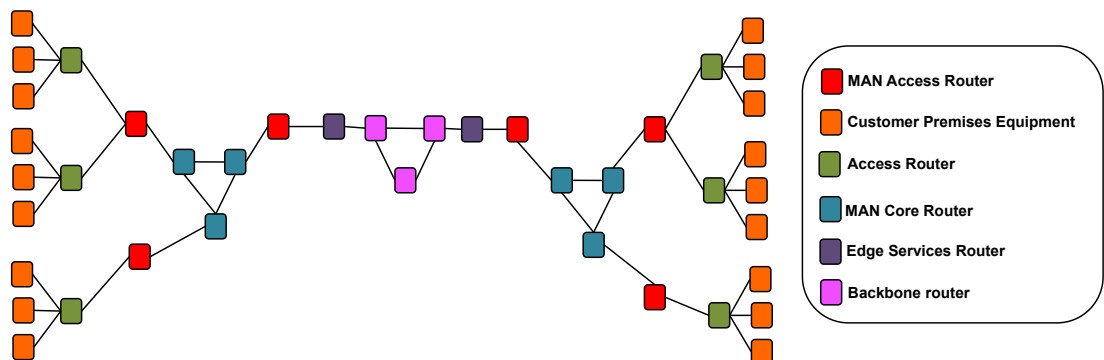

**Figure 10.** Physical systems in the multi-layer QoS experiment.

Each CPE router in the network is running an instance of the rina-echo-time server application and another instance from the rinaperf server application, accepting client flows from other CPE nodes. Each CPE node runs 4 rinaperf flows at 1 Mpbs and 4 rina-echo-time flows to all the other CPEs in the other MAN, each flow requesting a different level of quality. Hence each node is running 72 concurrent flows that result in 40 Mbps of traffic (16 ∗ 1 Mbps or rinaperf traffic and 4 Mbps of aggregated rina-echo-time traffic). In total the scenario features 1296 concurrent flows. Figure 11 depicts result of this experimental setup, showing the delay and loss measured by the 18 rina-echo-time flows at nodes CPE63 and CPE41 respectively (the flows in other nodes exhibit a very similar pattern). The Figure show flows grouped in two very differentiated sets: flows belonging to high urgency classes experience a relatively short end to end delay (around 20–30 ms), while flows belonging to low urgency classes experience a very high delay (around 1300 ms). The QTA scheduling policies effectively guarantee low latency for high urgency classes, but the delay experienced by low urgency classes is too high—even taking into account that the bottleneck links are highly loaded (this is due to the lack of performance optimizations in the IRATI implementation).

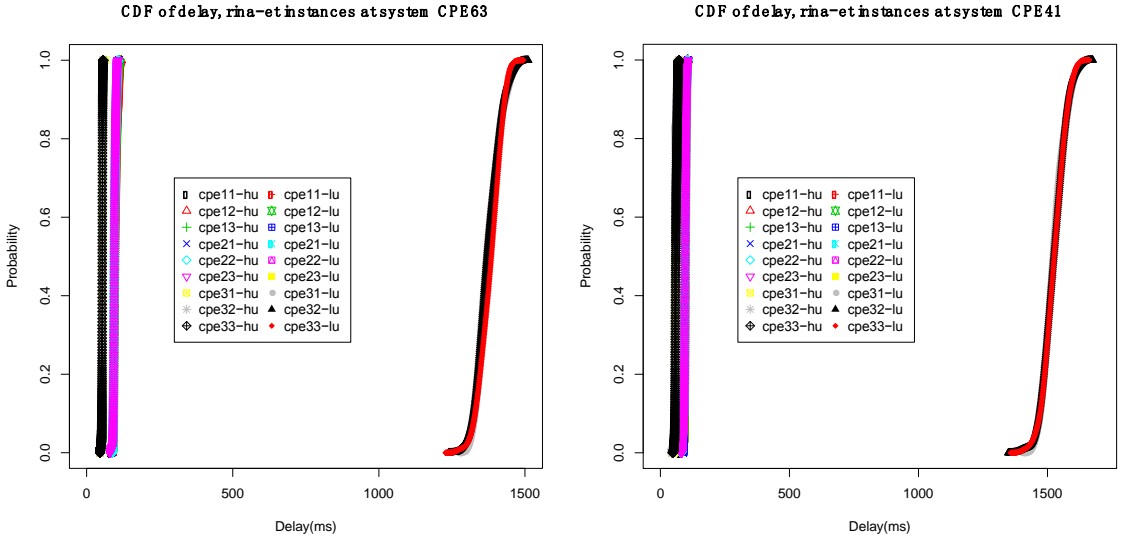

**Figure 11.** CDF of delay at systems CPE63 and CPE41, for multiple flows with different QoS requirements.

### 3.4. Distributed Mobility Management

The goal of this scenario is to validate the distributed mobility management [17] and multi-access capabilities [18] enabled by the layer structure and naming and addressing scheme of RINA. Part of these capabilities have already been illustrated in [7]: the Mobile Host—called User Equipment in this

experiment (UE)—exploited multi-homing via two WiFi interfaces to preserve service continuity during handover (a strategy usually called *make before break*). Multi-homing has been demonstrated both to two base stations of the same provider and to two base stations of different providers. This experiment features a handover between different providers that use different access technologies (wired and WiFi), and also illustrates the capability of routing the traffic of different applications through the DIFs of different providers according to policies configured at the UE. A detailed explanation of how to set up the experiment, including a bill of materials, configuration files, scripts and steps that need to be carried out, can be found at the IRATI tutorial#11 [19].

Figure 12 shows a view of the physical systems involved in the experiment. Three Raspberry Pis play the role of WiFi Access Routers (ARs) of the mobile provider network. The three ARs are connected to an Edge Router (ER), which is connected to a Core Router (CR). The CR connects the service provider network to the Internet via an ISP (Internet Service Provider) router. The ER is connected to a Data Centre (DC) gateway, which provides access to a small service-provider owner DC with two servers. The bottom of the figure shows a fixed provider network, that provides fixed connectivity to the customer edge router (CPE). The fixed provider network is connected to the Internet via ISP2. Both ISP1 and ISP2 are connected to ISP3, which provides access to Server3. All these systems (except for the Raspberry Pis) have been deployed as KVM/QEMU Virtual Machines (VMs).

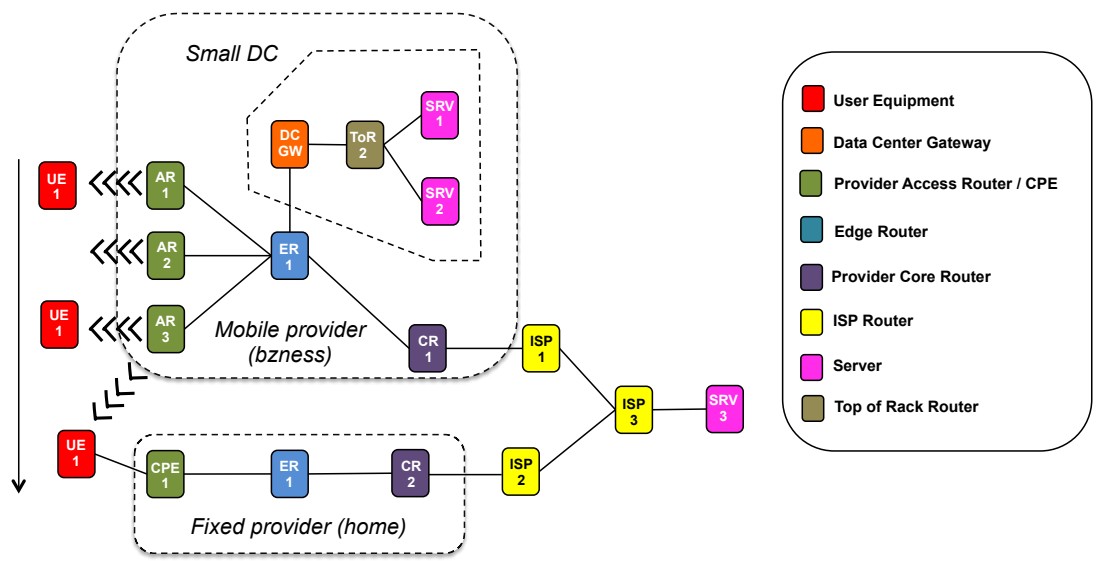

**Figure 12.** Physical systems present at the distributed mobility management experiment.

A single UE device (a laptop) starts the experiment connected to one of the mobile provider's access routers. The user with the UE device is accessing two applications: one located at the edge datacenter of the mobile provider, running at *Server 1* and reachable via the *Slice 1 DIF*; and the other one running at *Server 3* and reachable over the *Internet DIF*. The user with the UE is travelling home and roams through the mobile provider wireless access routers executing two handovers: *AR1* to *AR2* and *AR2* to *AR3*. When the user arrives home, it plugs its UE device to the cable that connects it to the fixed provider. The user has a policy configured in its UE device that, whenever the *Internet DIF* is available through the home provider, this one must be used. Hence, the UE device executes a handover and now it uses the wired interface provider by the home provider to reach the *Internet DIF*. However, it continues connected to the mobile provider via the WiFi interface, since the home provider does not have access to the *Slice 1 DIF*.

Figure 13 shows the organizations of IPC Process instances at the UE system, and how it evolves through the experiment lifetime. The UE has three shim IPC Processes: one to wrap each WiFi interface (*wlan0* and *wlan2*) and one to wrap *VLAN 40* on Ethernet interface *eth0* (this one is the fixed interface to connect to the home fixed network). As the UE moves, the shim IPCPs over the WiFi interfaces leave

and join different DIFs (over the different WiFi networks provided by AR1, AR2 and AR3 respectively). During this time, the IPCP belonging to the *Mobile Network DIF* (the blue one on Figure 13) changes its point of attachment to the different shim WiFi IPCPs, and different routing updates take place on the *Mobile Network DIF* to signal these changes. When the UE is plugged to the home network, it joins the *Fixed DIF* (orange IPCP in Figure 13), and the IPCP that belongs to the *Internet DIF* (green IPCP in Figure 13) changes its attachment from the *Mobile Network DIF* to the *Fixed DIF* (the same way the blue IPCP changed its point of attachment when the UE roamed through different WiFi networks). Applications supported by the *Internet DIF* do not notice the change.

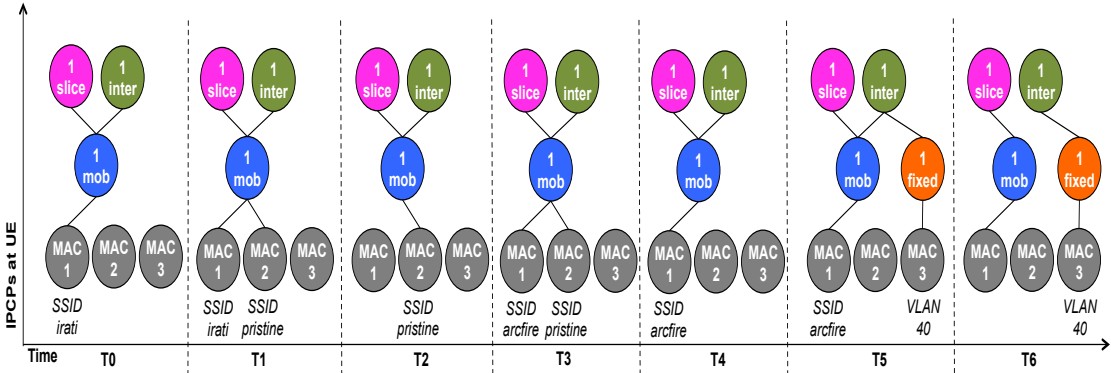

**Figure 13.** Physical systems present at the distributed mobility management experiment.

Service continuity is preserved through all this process, and in fact packet loss reported by both instances of the traffic generation applications running on top of the *Slice 1 DIF* and *Internet DIF* has been zero. Note that the handover procedure described in this experiment just leverages the inherent multi-homing capabilities of the RINA structure and link-state routing policies, no mobility-specific additions have been necessary. Hence, executing the handover procedure at the *Mobile Network DIF* layer or at the *Internet DIF* layer involves the same procedures. In larger scenarios there will be some differences and variations across layers depending on their size and operational environment, but nothing more than the usual policy differentiation (authentication, routing, forwarding) already foreseen in the RINA architecture.

## 4. Simplification of Network Management

A graphic representation of a RINA network is the layered view of the original design, as shown in Section 2, focusing on the DIF layers and hiding the nodes in the background. A RINA network management system needs to visualize each node in isolation, with its DIF structure, and the dependencies between the DIFs. A graphic representation that is better suited for this is called an onion diagram, shown in Figure 7 for an example network. The DIFs in the nodes are connected (using the same color as the respective DIF). These connections show how the communication between IPCPs of the same DIF happens between nodes, including which other DIFs and physical layer DIF it crosses. Onion diagrams can show a large amount of nodes in the same graphic or figure, which allows for rapid manual comparison of nodes. This representation is a good visualization of potentially complex node configurations with DIF structure, DIF dependencies, required configuration of them, and involved IPCPs (adding some dots for them).

One of the main insights during the ARCFIRE project was that the consequent application of RINA abstractions leads to significant improvements in configuration management, increasing performance of network management systems, decreasing management software complexity, and facilitating automation. In a network management system, a node can be created and validated independent of any other node, its configuration only requires its DIF and their connections (plus the DIF's QoS cubes). Management, especially performance management, will come down to how effective is congestion

management and resource allocation within a DIF. What is the loss rate due to congestion? What is the utilization of the graph? How often was it outside its QoS-cube? etc. This not only greatly simplifies management, but makes it much more quantifiable.

A RINA network can be created and validated using only individual node configurations as has been demonstrated in the Rumba tool. From the network (nodes) graph, a DIF graph can be generated, showing all defined DIFs and their connections, and a graph with point-to-point connections between nodes, or multi-point if they are physically connected to a broadcast medium (referred to as a physical connectivity graph). After validating these two graphs, the IPCP graph can be generated showing all IPCPs with their DIFs and their connections. These representations are shown for the example network of Figures 14 and 15.

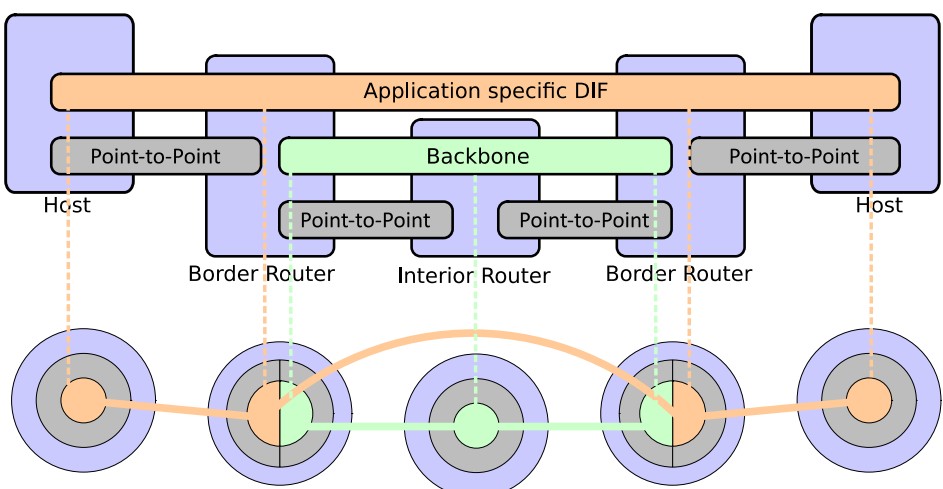

**Figure 14.** Visualizing a RINA network with an onion diagram.

A Distributed Management System (DMS) was created that uses this technique to validate a network [20]. Using the DMS, we have realized 50 test runs for 24 different networks on 6 different machines (spread over 3 different hardware platforms). Overall, we have run 6950 individual tests creating 945,400 nodes, 32,200 DIFs, 980,700 point-to-point DIFs, with 1,918,600 policy triggers. The number of created networks (or network segments) and the number of triggered DMS strategies is the same as the number of tests, 6950. All of this happened in a combined time of 55:42:05 h. In other words, we emulated creating the equivalent of 18.05% of all cells in the U.S (5,238,546) in under 2 days and 8 h. This means that this DMS strategy can handle configuration management for networks of extreme scale.

Table 2 summarizes the results of the individual tests carried out during the network management experiments, for each of the different measured KPIs (Key Performance Indicators).

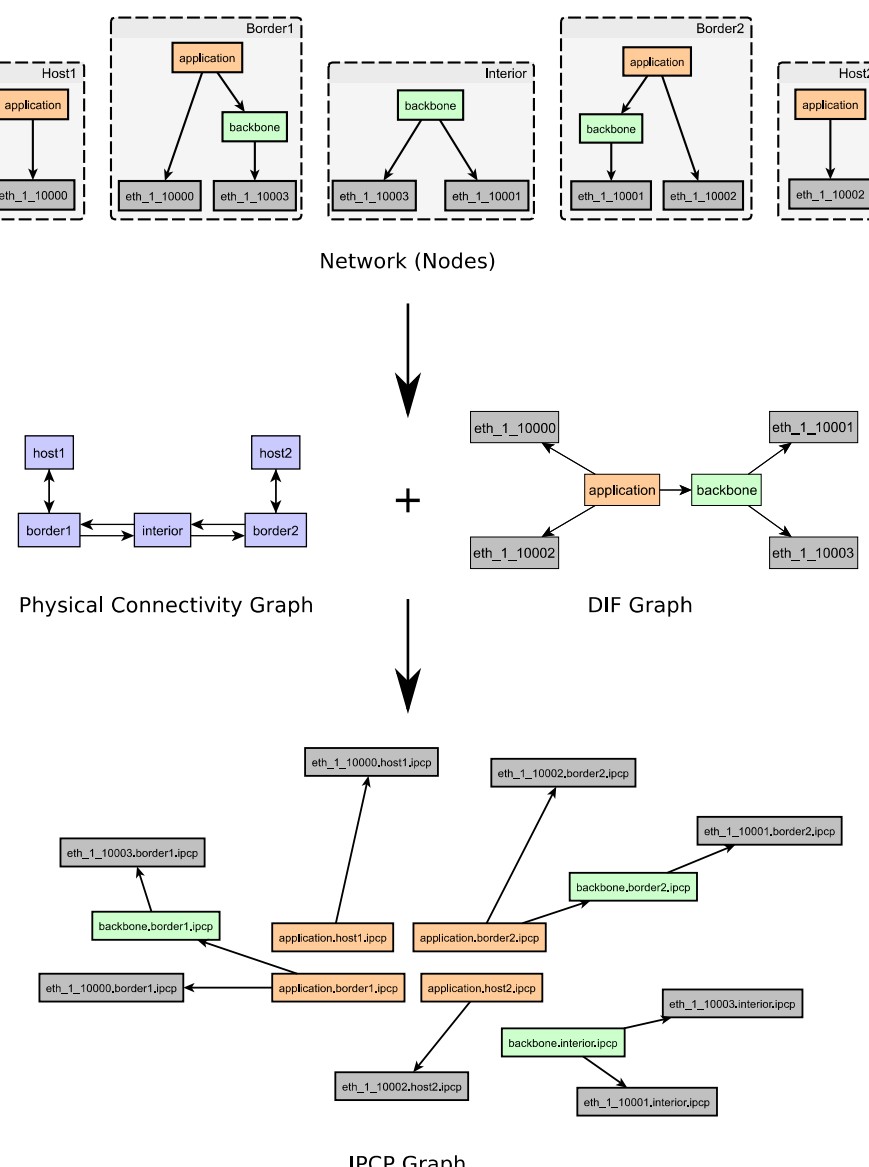

**Figure 15.** Validating the RINA network on Figure 12.

- Speed of node creation and validation. The average time to create or validate a node is 85 ms (on Raspberry Pi), 15 ms (on UNIX machines), and 3 ms (on Windows/-Cygwin). The creation adds a node to a network with its configuration. The validation runs a topological sort over the nodes' configuration, and adds its DIFs and their structure to the network. These times scale linear up to larger networks, non-linear for the very large metro networks.

- Complexity. We measured the complexity of nodes for the operations creation and validation. The minimum complexity for node creation is 5, for node validation is 3. The maximum complexity for node creation is 28, for node validation 48. The largest complexities can be found in the metro networks. The other networks have node creation complexities between 5 and 13, node validation between 3 and 12. From the numbers we can see that most nodes are rather simple in terms of DIFs and PtP DIFs, while being part of comprehensive and complex service provider networks. This simplicity is largely due to the RINA simplification using DIFs instead of complex TCP/IP configurations without scope separation.

- Degree of automation. The test runs are fully automated using DMS scenarios. However, we did experience race conditions on slow machines (in this experiment the Raspberry Pi) for all networks,

on medium size machines (in this experiment on Windows/Cygwin) for larger networks, and on all machines for very large networks. So, the degree of automation is 100% on most test runs on most machines, but less in the mentioned conditions.

- Touches. The number of manual operations for the management strategy to succeed is 0 on small to large networks on standard or powerful UNIX machines. The number increases by 1 per network for very large networks.

- DMS Scale, including time and cost. Taking each individual test run of each of the modelled networks as a network segment, we can see that the DMS is able to handle super large networks. In this context, we created a network with 6950 segments with overall 945,400 nodes in 55:42:05 hours. On all the 4 large UNIX machines combined, and without the very large metro networks, we created a network with 4000 segments and overall 57,600 nodes in 10:07:23 hours. We can state that scaling out the DMS components happens in less than 1 s, while scaling out APEX (the underlying DMS engine) can take 5 or 10 s, depending on the chosen option. This means that we can react to event storms within a few (minimum 5) seconds for the DMS strategy. This time 'should' be enough if we get an early indicator of an event storm, and the potential size the storm implies. Since event storms for configuration management are only when either a large network (many nodes) or a large number of smaller networks (with some or many nodes are created. These actions tend to happen in planned operations, not in unusual or accidental operations. One example is the action to add for instance 10,000 new nodes to a mobile network, or to deploy a new metro network, or to initially create an enterprise network.

**Table 2.** KPIs measured during the network management experiments. P = Raspberry Pi; U = large UNIX machine, C = Windows Cygwin machine.

| KPI | Part | Specialized KPI | Value | Description |
|---|---|---|---|---|
| Speed | DMS | Network | 4 min | Large, 50 tests |
| | Test runs | Create node | P: 85, U: 15, C: 3 | milliseconds |
| | | Validate node | P: 85, U: 15, C: 3 | milliseconds |
| Scale | Networks | Scale of networks | 2–2110 nodes | Small < 80, Large < 150 |
| | | | 207.300 nodes | Very large > 150 |
| | | | 945.400 nodes | Aggregated extra large |
| | Test runs | DMS | Very low | No requirements on scale |
| | | DMS, time and cost | Medium | New APEX distribution |
| Complexity | DMS | Lines of Code | 3019 | 2327 Bash, 375 Python |
| | Strategy | Lines of Code | All: 3072 Compact JS: 517 | JavaScript(JS): 916 APEX CLI: 840, AVRO: 316 |
| | Networks | Node Complexity | Create: 5–28 min/max (10–6180) | Min/max operations, ms |
| | | | Validate: 3–48 min/max (6–2037) | Min/max operations, ms |
| | | Network Complexity | 28 to 32468 | Tiny to very large |

**Table 2.** *Cont.*

| KPI | Part | Specialized KPI | Value | Description |
|---|---|---|---|---|
| | DMS | Scenario Automation | High but no 100% | Easy to get to 100% |
| | | Create Configuration | Very low | Automated with scalingtime |
| Degree of Automation | Networks | Create Network | 100% | Fully automated |
| | | Create Network Graphs | 100%, Medium | GraphML, post-processing |
| | Test runs | Complete a test run | Medium | Wait points (race conditions) |
| Touch | Networks | Manual operations | 0–1, 0–4 | Mandatory, optional |
| | Test runs | Manual operations | 122 in 139 networks | With better policies |

## 5. Discussion of Experimental Results

### 5.1. Resiliency Experiments

Different resiliency schemes are possible per layer, but in the experiment reported in Section 3.1 we focused on using Loop-Free Alternates (LFAs) in a single layer. We first showed that the recovery scheme functionally works, on a very simple setup, three nodes connected in a triangle. Then, we measured an upper bound for the recovery time. Fast failure recovery, i.e., below 50 ms, is definitely achievable. We recover in at most 13 ms. We then turned towards a bigger experiment, by emulating the GEANT network, which is deployed all over Europe. We measured the bandwidth before and after a link failure, for regular link state routing and when using LFAs. We showed that traffic is recovered in case of a failure, at the cost of higher bandwidth usage in the network. We also presented latency measurements of different flows. We see that after recovery, there is not only a higher bandwidth usage but also an increase in delay. All average round trip times stay below four milliseconds. Both observations are of course simply because more hops are used to reach the destination. We demonstrated that the experiment is very reproducible, largely thanks to Rumba [10].

### 5.2. Renumbering Experiments

Shortcomings in the naming and addressing structure of the current Internet protocol suite make network renumbering a tedious, error-prone and expensive procedure. The lack of application names causes the network to bind an application flow to an IP address and a transport layer port number. If the IP address of the source or the destination of the flow changes, the flow identity is lost and the flow is no longer usable. In contrast, the comprehensive naming scheme of RINA makes renumbering problems in IP networks non-issues and enables dynamic network renumbering. Flows are associations between application names, only locally bound to IPC Processes via a port-id. Addresses are just location-dependent synonyms of IPC Process names. The identity of IPC Processes is represented by their location-independent application name: authentication and access control operations are performed in terms of AP names, not IPCP addresses. Hence renumbering does not interfere with such procedures.

In the renumbering experiments we have analyzed how the complete naming and addressing architecture embodied by RINA allows RINA networks to be renumbered live, without significantly impacting the performance perceived by existing flows or impairing the ability to crate new ones. Renumbering a network with multiple layers today is a maintenance event: it has to be carefully planned, requires humans in the loop and it takes one or more days to complete. With RINA renumbering can be done life, dynamically, in a fully automated way and in a time period between tens or hundreds of milliseconds to a few seconds (depending on the network size).

This property can be applied to multiple use cases, such as: (i) *network address space consolidation*, merging two or more address spaces from different companies after an acquisition; (ii) *network address space optimisation*, changing the addressing policy of a DIF if it is no longer optimal due to changes in the DIF (e.g., the DIF has grown a lot since the addressing policy was designed, or its structure has changed); or (iii) *keeping addresses of mobile hosts aggregatable* as they move through different subnets.

### 5.3. Quality of Service Experiments

RINA provides a consistent QoS model from the application to the physical wire: applications that wish to do so can provide quality requirements to the network in a technology-independent way. There is no need to use Deep Packet Inspection (DPI) techniques to identify classes of traffic and infer the quality requirements of the application; hence applications doing end to end encryption can be supported with the appropriate level of quality. Layers (DIFs) provide QoS requirements to lower layers using the same abstract API, which means that there is no need to standardize QoS cube identifiers across DIFs (though the semantics of the QoS parameters passed through the layer API must be standardized).

QoS cubes provide a useful abstraction to communicate the ranges of quality supported by a DIF to its users, in a way that is decoupled from the mechanisms and policies used internally to enforce the quality expressed by each QoS cube. EFCP traffic marking enables a clear identification of PDUs belonging to different QoS cubes, and enables resource allocation policies (routing, scheduling, congestion control) to act consistently across a DIF [21]. QTAmux scheduling policies have proven a useful means to differentially allocate loss and delay to groups of flows, allowing DIFs to provide consistent levels of quality to the applications using them even at high loads. Moreover, the abstract QoS model provided by the deltaQ [22] theory allows an application to describe statistical bounds in the loss/delay required for its flows in a technology-independent way, which can be implemented in practice. The work started by ARCFIRE will allow future research to understand the join behavior of the QTAMux scheduling policies working in conjunction with ECN-based congestion management policies, giving the DIF the capability to limit the quality degradation experienced by lower urgency classes by decreasing the traffic load offered to the network.

### 5.4. Distributed Mobility Management Experiments

A mobility management solution requires two identifiers: one that doesn't change as the mobile host moves, so that communication endpoints keep stable identities, and on that does change, reflecting the position of the device in the network. Currently, the only identifier assigned to an entity in a Mobile Host is the IP address, but this single identifier cannot satisfy both requirements simultaneously.

RINA assigns location-independent application names and location-dependent addresses to IPCPs, resulting in mobility being supported without the need for home routers, foreign routers, tunnels, anchors, or specialized protocols. Managing mobility requires a combination of routing updates, changing addresses of IPCPs and designing the number and size of layers in different parts of the network to accommodate the load, scale, and rate of change of the devices to be supported. Mobile Hosts can roam through multiple provider networks and multiple access media without causing service disruption at the application level: service continuity is preserved as long as the destination applications are reachable [7]. Moreover, different applications may access different services from multiple providers, utilizing multiple underlying physical media simultaneously or as a fail-over.

### 5.5. Network Management Experiments

Results presented in Section 4 provide substantial experimental evidence that the consequent application of RINA abstractions leads to significant, we argue drastic, improvements in configuration management, especially for performance, management software complexity, and automation. The applied RINA abstractions are:

- The separation of mechanism and policy is the key abstraction, especially for network management. Mechanism' and 'policy' are relative and related to the scope of the system. In other words what is 'the mechanism' in one system (and scope) can be 'policy' in another (larger) system and scope.
- There is only one application protocol (CDAP, the Common Distributed Application Protocol) with only six operations: create, delete, read, write, start, and stop. CDAP operations are executed on well-defined objects in the Resource Information Base (RIB).
- Management is monitoring and repair, everything else is most likely control. The term management and its activities (often called management functions) are relative, i.e., more a continuum than a specific box in a system architecture. We can therefore use 'monitoring and repair' on different system levels (with associated scope).
- A network in the RINA sense is a set of nodes and DIFs plus the IPCPs in the DIFs. This seems to imply that we need more than one dimension to show and explore a Recursive InterNetworking Architecture (RINA) network, i.e., a simple 'topology' is not enough.

During the execution of the experiments we have identified and studied a number of further abstractions, namely:

- A node can be created and validated independent of any other node. Its configuration only requires its DIFs and their connections (plus the Distributed IPC Facility (DIF) QoS cubes). This has been demonstrated in the Rumba tool.
- A network can be created and validated only using individual node configurations. There is no further information required.
- All aspects of a RINA network can be shown and explored using four different graphs: a network (nodes) graph with the individual nodes and their configuration, a PtP graph with point-to-point connections between nodes, a DIF graph showing all defined DIFs and their connections, and a network graph (or IPCP graph) showing all IPCPs with their DIF and their connections. Some of these graphs are trees (in the mathematical sense), the PtP graph can be any type of graph.

## 6. Conclusions and Future Work

During the 7th Framework Programme and Horizon 2020, the European Commission made an investment into research into Future Internet with various projects on next-generation architectures. IRATI started a line of research into RINA, developing a prototype that was used by the GÉANT open call project IRINA and significantly extended by the PRISTINE project. These projects were necessarily heavy on development, with their most important outcomes proof-of-concept validation, qualitative insights, and quantitative evaluations comprising predominantly out of unit tests of the internal mechanisms with developed policies. Apart from the obvious effort in development, a very time-consuming aspect of these projects was deploying the prototype in the testbeds. For people new to the prototype, this presented an almost insurmountable barrier to engage in RINA research.

Using a CON scenario as a guide, ARCFIRE progressed the RINA software suite to the point that it is mature enough for large-scale deployments and long-lived experiments. The prototypes are now capable of running testbed deployments of over 100 nodes with thousands of active flows, providing stable and repeatable measurements between deployments. ARCFIRE developed the Rumba framework, which significantly reduces the time and effort needed to setup and execute experiments on FIRE+ and GENI. We foresee that the capacity of Rumba to significantly lower the barriers for RINA experimentation will make it an essential tool to increase the number of research organizations that take an active role in RINA research.

With these toys in hand, ARCFIRE took the first steps to build a body of quantitative evidence of some of the expected benefits of RINA networks. ARCFIRE showed that RINA networks are capable of delivering services that require fast failure recovery and guaranteed end-to-end QoS. Network renumbering is simplified and can be used to maximize the aggregation of addresses in routing

tables. The principles that underlay RINA lead to significant gains and simplifications for mobility management and configuration management.

From a network service provider perspective, ARCFIRE demonstrates that a step-by-step adoption of RINA using islands in different network segments can tackle painful networking issues without radically disrupting a service that has become critical to so many applications. This implies as well that a gradual transition of employed network engineers towards operating RINA networks is possible. While the uptake within the network service providers is still limited, ARCFIRE results have contributed to demonstrate the application of RINA as a viable option in many present and future scenarios. ARCFIRE results have been leveraged in a Telecom Infra Project (TIP) end-to-end slicing group collaboration with major vendors and operators [23], and also in the recently announced RINArmenia initiative [24].

Up to now most RINA research—including that of ARCFIRE—has focused on evaluating the properties of RINA at the architectural level, using relatively simple policies to exercise the architecture. Planned future work will be focused on designing concrete policies for specific use cases where RINA can bring the most benefits at the lowest adoption cost (e.g., large-scale application specific virtual networks, software-defined datacenters, network substrate for agile connectivity between Virtual Network Functions), and quantifying the benefits in real-world deployments. In order to enable such validations, there is also a need for higher performance RINA router implementations, which could exploit the current advances in programmable network technologies such as P4 [25].

**Author Contributions:** Conceptualization: E.G., S.V. and D.S. Experiments (software, execution and analysis): Resiliency: S.V., D.S., D.C., M.C., V.M. Renumbering, QoS and Mobility: E.G., M.T., D.L., J.D., L.C. Network Management: S.v.d.M. Writing—original draft preparation: S.V., E.G. Writing—review and editing: all authors. All authors have read and agreed to the published version of the manuscript.

**Funding:** This work is partly funded by the European Commission through the ARCFIRE project (Grant 687871), part of the Future Internet Research and Experimentation (FIRE) objective of the Eighth Framework Programme (Horizon 2020); and also by the AGAUR and the Catalan "Secretaria de Universitats I Recerca", through the "Grup de recerca preconsolidat" (2017 SGR 1658).

**Conflicts of Interest:** The authors declare no conflicts of interest.

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
