# Peer review of "ARCFIRE: Experimentation with the Recursive InterNetwork Architecture"

_computers, doi:10.3390/computers9030059_

Round 1

Reviewer 1 Report

Although we understand that any specific field may involve its own technical terms, there are still too many abbreviations appeared in this manuscript without their full names introduced first so that a reader not familiar with them would hardy realize the details. In addition, apart from the introduction to the ARCFIRE project, the work provides only a few quantitative results, which may not clearly confirm the argument given at the end of abstract that RINA can offer various simplifications in network management when compared to other technologies.  

Author Response

Dear Reviewer 1,

Thanks a lot for your comments, we have tried to address them in the revised manuscript.

REVIEWER's COMMENT #1:

"Although we understand that any specific field may involve its own technical terms, there are still too many abbreviations appeared in this manuscript without their full names introduced first so that a reader not familiar with them would hardy realize the details."

We have expanded all the acronyms in the text, and provided a short explanation when the text was not providing enough context to make the acronyms easy to understand. Also, in order to increase the clarity of Section II, we have added another Figure (current Figure I) and a table (Table I) providing a summary of architectural benefits of using RINA to design a converged operator network (which should also help people that has not read about RINA before).

REVIEWER's COMMENT #2:

"In addition, apart from the introduction to the ARCFIRE project, the work provides only a few quantitative results, which may not clearly confirm the argument given at the end of abstract that RINA can offer various simplifications in network management when compared to other technologies"

We have added an image to facilitate the understanding of the experiment described in section 3.1. On section 3.2 we have better described the implications of the results achieved in the experiment. On section 3.3 we have added a figure to describe the single-layer QoS experiment and enhanced the discussion of results. We have also added the descriptions and results from a second QoS experiment, this time with multiple layers. On section 4 we have added a table summarizing all the numerical results of the network management experiment, as well as a discussion of the KPIs measured in such experiments.

Reviewer 2 Report

This paper reports on the experiments carried out on a technology-edge implementation of a novel network architecture based on nested flows and free stacking. The focus is on the scalability of the testbed for supporting large, real networks. As a project-description paper, this manuscript is very well written and presented, it is clear, with high-quality illustrations and diagrams. The main ideas are transmitted effectively and the general picture is shown in a neat form to the readers.

While the concepts and ideas used in RINA (the novel architecture) can be traced back and found in other architectural proposals for networks, the work presented in the paper is clearly innovative and shows a promising direction for advancing the network technology in the forthcoming years.

Therefore, as a tutorial paper showcasing a novel technological approach, the manuscript is suitable for publication.

Author Response

Dear Reviewer 2,

Thanks a lot for your feedback. By your comments we understand that you are not requesting any changes to the manuscript.

Reviewer 3 Report

  • Is this the final version of paper? The authors submitted version with track changes ON, it looks like pre-final version.
  • Abstract should provide more details about paper
  • Tables should be approprirately formatted
  • New section with summary of the achieved results should be introduced, i.e. section 5. Discussion
  • The paper looks more like project report than scientific paper
  • Literature: you need to include more related papers, the most of the papers are deliverables and authors work on other conferences
  • Future work is missing

Author Response

Dear Reviewer,

Thanks for your feedback! We have applied the following changes:

  • Disabled track changes
  • Enhanced the abstract
  • Converted Table 2 to the proper format
  • Added section 5 with a discussion of the results
  • Added paragraph on Future work in section 6
  • Added 8 more references

Best regards,

Eduard

Round 2

Reviewer 1 Report

The writing may be still improved to increase its readability.

Author Response

Dear reviewer,

Thanks for your comments!

We have added a few more content (mainly section 5 and larger description of experiment 3.4) and improved abstract and conclusions.

Best regards,

Eduard

Reviewer 3 Report

All my comments are addressed in details with high  quality contributions.